# An Overview of Cotton Gland Development and Its Transcriptional Regulation

**DOI:** 10.3390/ijms23094892

**Published:** 2022-04-28

**Authors:** Masood Jan, Zhixin Liu, Chenxi Guo, Yaping Zhou, Xuwu Sun

**Affiliations:** 1State Key Laboratory of Cotton Biology, School of Life Sciences, Henan University, 85 Minglun Street, Kaifeng 475001, China; masoodjan@henu.edu.cn (M.J.); zxlsch2019@163.com (Z.L.); chenxi1445@163.com (C.G.); zhouyaping@henu.edu.cn (Y.Z.); 2State Key Laboratory of Crop Stress Adaptation and Improvement, School of Life Sciences, Henan University, 85 Minglun Street, Kaifeng 475001, China; 3Key Laboratory of Plant Stress Biology, School of Life Sciences, Henan University, 85 Minglun Street, Kaifeng 475001, China

**Keywords:** cotton (*Gossypium* spp.), gland development, gossypol gland regulation, molecular mechanism, transcriptional regulation

## Abstract

Cotton refers to species in the genus *Gossypium* that bear spinnable seed coat fibers. A total of 50 species in the genus *Gossypium* have been described to date. Of these, only four species, viz. *Gossypium, hirsutum, G. barbadense, G. arboretum*, and *G. herbaceum* are cultivated; the rest are wild. The black dot-like structures on the surfaces of cotton organs or tissues, such as the leaves, stem, calyx, bracts, and boll surface, are called gossypol glands or pigment glands, which store terpenoid aldehydes, including gossypol. The cotton (*Gossypium hirsutum*) pigment gland is a distinctive structure that stores gossypol and its derivatives. It provides an ideal system for studying cell differentiation and organogenesis. However, only a few genes involved in the process of gland formation have been identified to date, and the molecular mechanisms underlying gland initiation remain unclear. The terpenoid aldehydes in the lysigenous glands of *Gossypium* species are important secondary phytoalexins (with gossypol being the most important) and one of the main defenses of plants against pests and diseases. Here, we review recent research on the development of gossypol glands in *Gossypium* species, the regulation of the terpenoid aldehyde biosynthesis pathway, discoveries from genetic engineering studies, and future research directions.

## 1. Introduction

Cotton plants in the genus *Gossypium* possess pigment glands, which are often referred to as “gossypol glands”. These gossypol glands appear as black or brownish-red dots in all the tissues of cotton plants, with the exception of pollen and the seed coat [1,2]. Gossypol is a unique secondary metabolite that can enhance the resistance of cotton plants to pests and diseases [3]. The toxicity of cottonseeds to humans and ruminants due to the presence of gossypol limits their use as a source of protein and oil [4,5]. Several previous studies have examined the genesis and development of gossypol glands in cotton and the production and accumulation of gossypol during gossypol gland development [6,7]. The gossypol glands of cotton plants emerge from a mass of meristematic cells as a group of secretory cells, and mature gossypol glands consist of a subcuticular storage cavity surrounded by one to three layers of secretory cells [8]. The mechanism of gossypol gland formation involves the dissolution of gossypol gland cells into gland primordium, followed by the degradation of the gland primordium into the gossypol gland cavity [9]. Several materials produced at different points throughout this process mediate the degradation of the gossypol gland primordium and the development of the gossypol gland cavity [10]. Previous studies in cotton have shown that programmed cell death (PCD) plays a key role in the production of gossypol glands [11]. PCD of gossypol gland cells promotes the production and accumulation of gossypol; the products of the PCD of gossypol glands might thus contribute to the production of gossypol. Gossypol and its derivatives, as well as other secondary metabolic products in the gossypol gland, are the main chemicals that are stored in the gossypol gland and confer its color [12,13]. Cotton plants possess internal and external glands. Nectar glands are external glands that are often present on the skin, whereas internal glands (radius of 0.1–0.4 mm) are oval and spherical and are either black, brilliant yellow (orange), yellow-brown (yellowish-brown), green (red-brown), or purple depending on the species [5,7,14]. The glands of cottonseeds contain mostly gossypol with traces of deoxyhemigossypol, and the glands of cotton leaves contain hemigossypolone, a gossypol derivative [15,16], and heliocides. Gossypol is a triterpenoid aldehyde that has been used in medicine for its anti-tumor and anti-carcinogenic activities. It has also been used as an antifertility agent for male patients; as a pesticide to combat insect, fungal, and bacterial pests, and as an ingredient in various cosmetic products [17,18]. 

## 2. Developmental Changes in the Morphology of the Gossypol Gland in Cotton Plants 

Gossypol glands are present in all tissues of cotton plants, with the exception of the pollen and seed coat. Gossypol glands in older plants are also present in the phloem rays of the bark [19,20,21]. The number of gossypol glands is particularly high in *Gossypium barbadense*. The density and size of gossypol glands vary in different parts of cotton plants as well as among species and races. The gossypol glands of *G. barbadense* are darker and more conspicuous compared with those in other cultivated species. To characterize developmental changes in the morphology of the gossypol gland in cotton plants, we studied the microstructure of the gossypol gland in cotyledons at 2, 12, 36, and 72 h and embryos at 22, 25, 29, 35, and 50 days post-anthesis (DPA) that were collected and fixed in formaldehyde alcohol acetic acid from the near-isogenic line GI1 (*G. barbadense* with gossypol glands in both the seeds and plants). The tissues were dehydrated through a graded series of ethanol, embedded in paraffin, and cut into sections. The histological structure of the gossypol glands was characterized using a scanning electron microscope [22,23]. Several changes in the gossypol glands on the surface of the cotyledon of GI1 occurred during embryo formation. At 22 and 25 DPA, the gossypol gland was absent on the surface of the cotyledon (Figure 1A,B); at 29 DPA, the gossypol gland on the surface of the cotyledon was black (Figure 1C); at 30 DPA, the gossypol gland was light brownish-red (Figure 1D); and at 35 to 50 DPA, the amount of gossypol in the pigment gland cavity continued to grow, nearly filling the entire pigment gland cavity. (Figure 1E,F). 

The morphological changes of the gossypol glands during embryonic development were also characterized (Figure 1G–I). At 22 DPA, the cotyledon cells were closely arranged, and no special cells were present (Figure 1G). The gossypol gland primordium began to form around 25 DPA, and it consisted of a dozen cells with a black hue and a thick protoplasm that formed a ring around the gland. The gland primordium was spherical, and its cells were densely packed together in two to three layers on the surface. There were typically two to three large central cells with a visible cell wall and nucleus [2,6]. There were usually one to two peripheral cells, which were smaller than the central cells. The peripheral cells were compressed into long ovals, and the central cells were round (Figure 1H). Signs of differentiation appeared at 29 DPA, and a portion of the gland primordia tissue disintegrated into the gossypol gland cavity. Irregularities were observed in the core cells of the gland primordia that would later dissolve. When they began to dissolve, the distinction between the cell wall and the nucleus became less pronounced. The peripheral cells were compressed into a thin strip, which began to generate a small cavity (Figure 1I). The fuzzy appearance of the cells in the gossypol gland cavity stemmed from the destruction of the nuclei of gland primordia cells. Staining revealed a deep and thick wall, which might have developed due to the cell debris that accumulated following the degradation of gland primordia tissue. Filamentous material was present in the cavity because the peripheral cells had been compressed into long strips (Figure 1J). The disintegration of the core cells continued between 30 and 50 DPA, and the width of the gossypol gland cavity increased during this period (Figure 1K). 

At 50 DPA, a developing secretory cavity was visible in the center of the gossypol gland [19,21,24], and it was surrounded by several disintegrating secretory cells (Figure 1L). The microstructure of the gossypol gland during seed germination was also characterized. Two h after seed germination, the gossypol gland cavity on the surface of the cotyledon was visible, and it consisted of two to three layers of cells. The peripheral cells resembled long strips due to the extrusion process (Figure 1M). The long, extended morphology of the peripheral and central cells of the gossypol gland cavity was maintained between 12 to 36 h after seed germination. The barrier between the cell nucleus and the cell wall was blurry, and some filamentous material was present in the gossypol gland cavity (Figure 1N,O). The quantity of filamentous material in the gossypol gland cavity was higher 72 h after seed germination, and the color of this material darkened. The diameter of the gossypol gland cavity changed slightly during the germination stage (Figure 1P). The major cause of the color change in the gossypol gland between seed germination and embryo formation was the difference in the staining time between these two stages of development [6,15,19,22].

## 3. Functions of the Gossypol Gland in Cotton Plants

The growth of the global human population will require the increased production of food, fiber, and feed. However, crop production is limited by climate conditions, including the increased drought frequency and salinity in agricultural lands [5,25,26]. Cotton is an important crop for the production of fiber and food, and a total of 49 species belong to the genus Gossypium L. Cottonseed, soybeans, and rapeseed account for 6.9% of the world’s production of protein meal [27]. Cottonseed production on a global scale can potentially provide the yearly protein requirements for half a billion people, as cotton plants generate 1.65 kg of seed for every kg of fiber produced [28]. However, all cotton species possess lysigenous glands that produce terpenoid aldehydes, and the most dangerous is the sesquiterpenoid gossypol, which is toxic to both nonruminant animals and humans [29]. These gossypol glands store terpenoid aldehydes, such as gossypol, on the surfaces of cotton organs and tissues [2,28,30]. The structure of gossypol was first identified by Adams et al. (1938); cotton plants make gossypol, which is a phytoalexin. These glands increase the resistance of plants to pests and diseases [31,32]. 

Gossypol has been shown to have anti-carcinogenic, anti-HIV, and antibacterial activities and reduce fertility in males in vitro [33,34]. However, the full nutritional potential of the protein and oil of cottonseed has not been exploited due to its toxicity; gossypol also discolors cottonseed oil [35]. Cotton varieties in which seeds and plants lack gossypol glands have been developed, and these varieties either possess no gossypol or have extremely low gossypol levels [29]. Because the protein and oil derived from glandless seeds are free of gossypol and thus suitable for direct consumption [25], the resistance of these glandless cotton varieties to pests and the yield of cotton fibers are reduced; thus, glandless cotton varieties have not been widely cultivated [7,36].

Cotton has long been thought to be a crop that could provide a valuable source of fiber and food. Future breeding efforts are needed to develop cotton varieties that possess gossypol glands inside the roots, leaves, and stems (which aid the resistance of cotton plants to pests and diseases) and lack glands in the seeds to permit their safe consumption [37]. Alternatively, an ideal cotton cultivar might be characterized by delayed gossypol production until germination, which is a trait that has only been observed in a few Australian *Gossypium* species [7,36,38]. Many molecular biology and genetic engineering studies have examined the relationship between gossypol glands and gossypol production, and this work has led to several new findings that will have a major effect on the breeding and planting of cotton, the industrial processing of cottonseed, and even its use as animal feed and human food. These discoveries will greatly aid the development of agriculture and the economy and ensure global food security [2,5,31,39].

## 4. The Gossypol Biosynthetic Pathway 

Cotton plants produce a group of lineage-specific sesquiterpenoids, such as gossypol and hemigossypolone, that have antifungal, antibacterial, or insecticidal activity against a variety of herbivores, including the lepidopterans cotton bollworm and beet armyworm [40,41,42]. Gossypol is the major, if not only, sesquiterpene phytoalexin present in cotton seeds, and hemigossypolone is more abundant than gossypol in leaves [43,44]. Terpenoid aldehydes are produced in the lysigenous glands of cotton plants [45,46]. Gossypol is the main substance in the glands of *Gossypium hirsutum* in achlorophyllous plant sections; by contrast, gossypol methyl and dimethyl ethers are the most common substances in the glands of *G. barbadense*. Hemigossypolone is the major terpenoid aldehyde in the glands of the immature green tissues of *G. hirsutum*, and a novel quinone, hemigossypolone-7-methyl ether, has been identified in *G. barbadense* [15,47]. There is substantial variation in terpenoid quinones and their helicoid derivatives in wild *Gossypium* spp. and allied Gossypieae taxa. Numerous cadinene sesquiterpenoids and helicoids (sesterterpenoids) involved in disease and insect resistance are present in the gossypol glands of cotton plants [23,48,49].

Gossypol was originally thought to be produced from acetate through the isoprenoid pathway based on its structure. Previous studies have investigated the incorporation of mevalonate-2-14C into gossypol, which is a key step in the isoprenoid pathway, as well as the distribution of radioactivity in gossypol (Figure 2) [5,50,51]. Cotton terpenoid aldehydes and cadalene derivatives are sesquiterpenes (C15) generated by terpenoid metabolism in the cytosol through the mevalonate (MVA) pathway [36,52]. Farnesyl diphosphate (FPP) is converted into the linear carbon skeleton of sesquiterpenes in cotton [53]. FPP is cyclized by numerous sesquiterpene synthases to generate the molecular framework for distinct types of sesquiterpenes [54]. In cottonseed, gossypol is the main sesquiterpenoid produced, and the concentrations of desoxyhemigossypol (dHG) and hemigossypol are low. Hemigossypolone is produced from dHG in cotton leaves [48,55]. The cadinene enzyme, which is a soluble hydrophobic monomer with a molecular mass of 64 to 65 kD, has been isolated from a glandless cotton mutant [53,56]. Various cadinene sesquiterpenoids and helicoids (sesterterpenoids) are present in the gossypol glands of cotton and contribute to the resistance of cotton plants to disease and insect pests [39]. Figure 2 illustrates a possible mechanism by which these chemicals are produced. Infection of cotton stele tissue with *Verticillium dahliae* conidia increases the abundance of 3-hydroxy-3-methylglutaryl-CoA reductase (HMGR) mRNA and HMGR activity, suggesting that HMGR plays a key role in the production of sesquiterpenoids [33]. The enzymatic product of E via E-FDP cyclization in cotton extracts was later shown to be (+) cadinene (CDN) [53,57,58]. The enzymatic mechanism by which CDN synthase generates the cadinene structure of cotton sesquiterpenoids has been shown to involve the isomerization of FDP to a nerolidyl intermediate; cyclization to a *cis*-germacradienyl cation; a 1, 3-hydride shift; cyclization to a cadinanyl cation; and deprotonation to form CDN [33]. At a branch point in the MVA pathway, CDN synthase catalyzes the final step in the production of cadinene sesquiterpenoids from FDP. The cycloaddition of myrcene or ocimene to hemigossypolone to generate heliocides then occurs through a Diels–Alder reaction; monoterpenes and their precursors are produced in plastids through the 1-deoxy-5-xylulose-5-phosphate pathway (Figure 2) [5,15,50].

Many studies have examined the biosynthesis of gossypol and its derivates in recent years. One of the precursors to hemigossypol in *G. hirsutum* is 8-hydroxy (+) d-cadinene [59]. The cytochrome P450 monooxygenase *CYP706B1*, a (+) d-cadinene-8-hydroxylase involved in cotton sesquiterpene biosynthesis, is expressed in the aerial tissues of glanded cotton cultivars but is not expressed or expressed at an extremely low level in the aerial tissues of a glandless cultivar. The expression pattern of *CYP706B1* and the site at which CYP706B1 hydroxylates (+) d-cadinene indicate that *CYP706B1* functions in an early stage of gossypol biosynthesis and thus that *CYP706B1* could be a target for genetically engineering cotton plants to produce cottonseeds with higher gossypol levels [34,60,61]. Desoxyhemigossypol plays an important role in the production of these chemicals. A methyltransferase (*S*-adenosyl-L-met:desoxyhemigossypol-6-*O*-methyltransferase) has been identified, purified, and described in cotton stele tissue infected with *V. dahliae*. Desoxyhemigossypol-6-methyl ether is used to synthesize methylated hemigossypol, gossypol, hemigossypolone, or heliocides (Figure 2) [5,48,51].

## 5. Molecular Cloning of Genes Associated with Gossypol Synthesis and Gossypol Glands

Various genes involved in the terpenoid biosynthesis pathway have been cloned. Cadinene genes were initially cloned and functionally characterized from the A-genome of diploid cotton *Gossypium arboreum*; they are now part of a large multigene family in cotton [33,34], similar to the terpene cyclase genes that have been identified in other plants [51,62]. Numerous allelic and gene family variants of cadinene genes have been identified from both *G. arboreum* [33,49,63] and the allotetraploid (A + D genomes) *G. hirsutum*. The activity of cadinene enzymes and the expression of cadinene transcripts are induced in cotton stems inoculated with *V. dahliae* [34,56], as well as suspensor cultures of cotton treated with *V. dahliae* elicitors [57]. Cadinene appears to be controlled by the same genes during the development of these two cotton species. The expression of these genes increases during seed development and is linked to the production and deposition of gossypol in the lysigenous glands of the embryo [48,64,65]. The cadinene gene family in *Gossypium* has been proposed to comprise two main subfamilies, *cdn1-A* and *cdn1-C*, based on sequence similarity and differences in transcriptional control [33]. The structure of cadinene genes, including the number, location, and size of the exons and introns, is highly conserved, which is consistent with the genomic clones of other terpene cyclase genes, such as tobacco (5-epi-aristolochene synthase) (*Nicotiana tabacum* L.). Janga et al. studied the genes regulating gland development in cotton plants and sequenced approximately 2 kb of the promoter regions from each genomic clone; they found low sequence conservation between the promoter regions. Isolated areas of similarity were concentrated around the TATA box and transcription start site [51,63]. Luo et al. cloned and determined the function of (+)delta-cadinene-8-hydroxylase, a cotton sesquiterpene biosynthetic cytochrome P450 monooxygenase; they also cloned a 1.9-kb P450 that encodes a 522-amino acid protein with 48% similarity to soybean cytochrome P450 82 A3 and with a consensus heme-binding motif and an oxygen-binding pocket sequence [49,53,57]. This P450 is present in the leaves of the glanded cotton species *G. hirsutum* but not in the leaves of glandless cotton. The gossypol glands and terpenoids associated with gossypol glands are absent from the leaves of glandless plants. This indicates that this P450 enzyme is involved in the biosynthesis of terpenoids in cotton [60]. Suppressive subtractive hybridization and other approaches have been used to generate a subtractive library and a normalized cDNA library from a cotton mutant, Xiangmian 18, of gland morphogenesis-related genes. Some important genes have been cloned, such as the gene encoding the RanBP2 zinc finger protein in upland cotton [49,66]. Furthermore, the genes encoding *G. barbadense* desoxyhemigossypol-6-*O*-methyltransferase and the gene encoding cytochrome P450 associated with gossypol glands have been cloned and studied [16,53,63,67]. 

## 6. Transcriptional Regulation of Cotton Gland Morphogenesis and Pigmentation by *CGP1*, *GoPGF*, and *CGFs*

Given the importance of cotton as a fiber crop, understanding the development of gossypol glands and the synthesis of secondary metabolites, as well as how they can be used to improve the production and quality of cotton, have been major goals of current research [2,31,38,68]. However, the challenges associated with generating novel glandless mutants and cloning related genes using map-based cloning methods have impeded research progress. The cloning and characterization of *GoPGF* [68,69], as well as other *CGF* genes, in recent studies have provided important new insights into the formation of gossypol glands. *GoPGF/CGF3* regulates both gland morphogenesis and gossypol synthesis, *CGF1* plays a role similar to *GoPGF/CGF3*, and *CGF2* regulates the density of gossypol glands. *GoPGF/CGF3* also regulates gossypol synthesis and production (Figure 3). The silencing of *GoPGF* leads to the cessation of gossypol gland development in cotton, which results in negligible gossypol levels. Although preliminary observations suggest that the silencing and knockout of *CGP1* in glanded cotton produces a phenotype similar to that of the *gopgf* mutant, detailed analyses have revealed that *cgp1* mutants possess normally structured gossypol glands and numbers of gossypol glands similar to wild-type plants, which indicates that *CGP1* does not play a role in gland morphogenesis. The absence of glands in *cgp1* plants is the most likely explanation for their lack of colored pigments. The deletion of *CGP1* results in the down-regulation of many gossypol biosynthesis genes as well as a significant decrease in gossypol levels [2,31,45,48]. The development of gossypol glands does not appear to depend on gossypol production given that transgenic cotton lines with low gossypol levels (induced via the silencing of the key gossypol biosynthesis gene *CYP706B1*) show normal gland growth [68]. Gao et. al. showed that *GoPGF* controls both gland morphogenesis and gossypol production independently by binding to the promoters of WRKYs and terpene synthases (TPSs), respectively. The MYB transcription factor CGP1 regulates gossypol accumulation but not gland development, and it possesses transcriptional activity and interacts with GoPGF in the nucleus [45,48,49]. MYB proteins tend to form homodimers and heterodimers, which increases their affinity and specificity for DNA binding [57,69,70]. Thus, CGP1 and GoPGF might form heterodimers to regulate gossypol synthesis as well as the production of other terpenoids; however, they do not form heterodimers to regulate glandular growth (Figure 3). Although GoPGF is highly expressed throughout cotton plants, the expression of CGP1 in the roots is low, indicating that GoPGF might form homodimers or dimers with other transcription factors in the roots. The G-box motif in the promoters of several WRKY and TPS genes has been shown by yeast one-hybrid assays to be a binding site for GoPGF [38,68,71]. Whether the presence of *CGP1* increases the affinity of GoPGF to the promoters of WRKY and TPS genes or the in vivo transcription activation of target genes requires further study. Although the knockout of *GoPGF* results in the complete absence of gossypol, *cgp1* mutants possess some residual gossypol, which suggests that *CGP1* plays an important, but not essential, role in the regulation of gossypol synthesis. In addition to gossypol, several secondary metabolites are present exclusively in gossypol glands and confer their characteristic intense color [31,53].

## 7. Challenges, Conclusions, and Future Directions 

Ensuring food security has long been a major goal for mankind, and this is being challenged by human population growth and the increasing scarcity of arable land. There is thus a pressing need to develop ways to more efficiently utilize cotton plants as a source of fiber and food. Over the past few years, research on the gossypol glands and gossypol in cotton has focused on the production of glandless seeds and glanded foliage using molecular cloning and genetic engineering techniques so that cottonseeds can be directly consumed. [72,73]. The long-term goal of future research should be to understand the mechanisms by which genes control gland development and gossypol synthesis, as well as enhance the resistance of cotton plants to pests and pathogens, as this will increase the utility and economic value of cottonseeds. The development of cotton plants lacking glands in the seeds and leaves will have a substantial effect on the breeding, cultivation, and consumption of cotton. In the future, cotton will become a more valuable source of fiber, food, and oil, which will increase the world’s food security. Previous research has also enhanced our understanding of the roles of secondary compounds in plant tissues and the molecular mechanisms that control them. For example, the roles of artemisinin, a substance with antimalarial properties, in southernwood plants are similar to those of gossypol in cotton, and they are both stored in specific glands present in various tissues. Clarifying the molecular mechanisms underlying the synthesis of useful secondary compounds will benefit both farmers and the general public.

## Figures and Tables

**Figure 1 ijms-23-04892-f001:**
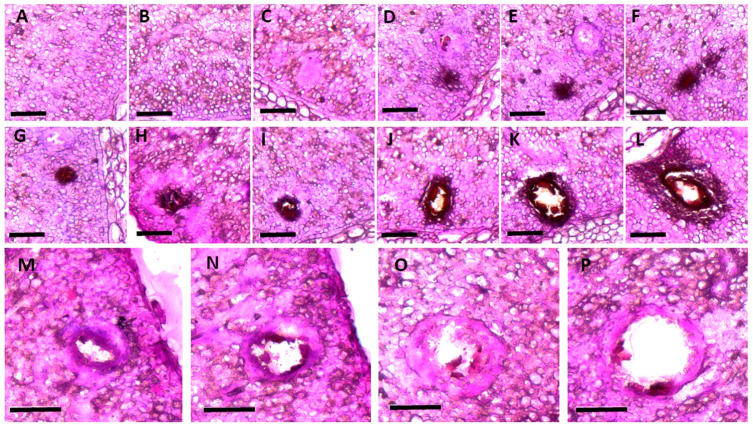
Developmental changes in the morphology of the gossypol gland during embryo formation. Gossypol gland morphology at (**A**) 22 DPA, (**B**) 25 DPA, (**C**) 29 DPA, (**D**) 30 DPA, (**E**) 35 DPA, and (**F**) 50 DPA. (**G**) Cell arrangement at 22 DPA. (**H**) Gossypol gland primordium formation at 25 DPA. (**I**) At 29 DPA, the primordium of the gossypol gland begins to disintegrate. Gossypol gland cavity at (**J**) 29 DPA, (**K**) 35 DPA, and (**L**) 50 DPA. (**M**–**P**) Gossypol gland formation during seed germination. (**M**) Gossypol gland cavity after 2 h, (**N**) 12 h, (**O**) 36 h, and (**P**) 72 h. Scale bar: 100 μm.

**Figure 2 ijms-23-04892-f002:**
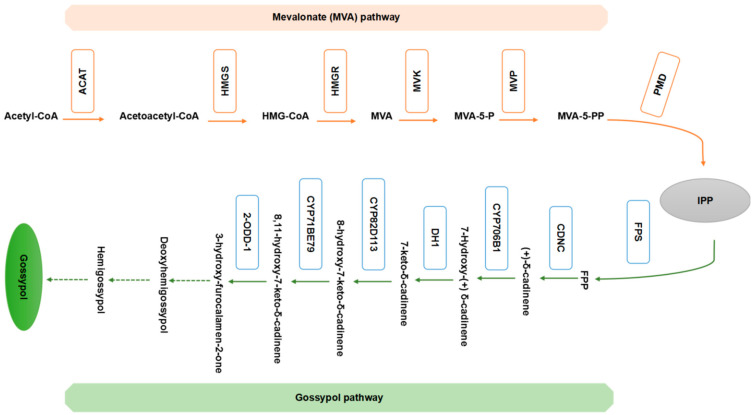
Genes of the mevalonate (MVA) pathway and gossypol pathway and their enzymes. Genes that contribute to the synthesis of gossypol and the enzymes that regulate their expression are presented for each step of the MVA pathway and gossypol pathway. Genes encoding identified enzymes or having expression patterns connected to gossypol biosynthesis are shown at the top. The enzymes involved in the MVA pathway and gossypol pathway, and intermediates include ACAT, acyl CoA-cholesterol acyltransferase; DMAPP, dimethylallyl diphosphate; FPS, FPP synthase; HMGR, HMG-CoA reductase; HMGS, 3-hydroxy-3-methylglutaryl-coenzyme-A (HMG-CoA) synthase; MVK, mevalonate kinase; MVP, phosphomevalonate kinase; PMD, diphosphomevalonate decarboxylase; and IPP, isopentenyl diphosphate. Different enzymes then catalyze the conversion of (+)-δ-cadinene, 7-hydroxy-(+)-δ-cadinene, and 7-keto-δ-cadinene to gossypol. Dashed arrows indicate unidentified reactions of the gossypol pathway.

**Figure 3 ijms-23-04892-f003:**
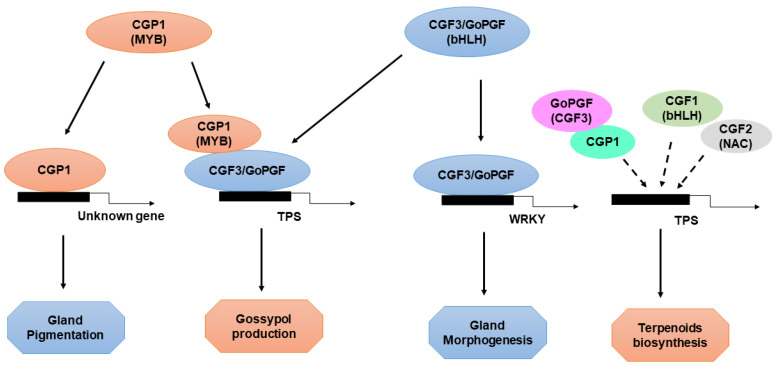
Regulation of gland formation and gossypol biosynthesis. The regulation of gland formation and gossypol biosynthesis. GoPGF protein, as a master regulator, controls the specification and differentiation of gland cells by regulating the expression of downstream genes. CGF2, a NAC transcription factor, also plays an important role in gland development and terpenoid biosynthesis. Moreover, GoPGF interacts with CGP1, an R2-R3 MYB transcription factor, to regulate the biosynthesis pathway of gossypol.

## Data Availability

All data supporting the findings of this study are available within the paper published online.

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
