# Peer review of "An Overview of Cotton Gland Development and Its Transcriptional Regulation"

_ijms, 2022, doi:10.3390/ijms23094892_

Round 1

Reviewer 1 Report

I reviewed the article titled: "An overview of cotton gland development and its transcriptional regula" and I found it interesting and, without few mistakes, well-prepared. The review provided by the authors to a large extent relates to the current state of knowledge. Knowledge compilation prepared by this research group can be used in far-reaching future research, which I find very useful and significant. Based on my experience, I recommend this article for publication in IJMS-journal after minor revision. Please check and verify the following points:

Introduction:

[33-37] The authors should restructure first sentence. As it is now, it seems that cotton has both pigment glands and gossypol. Third sentence starts from stating that this particular pigment glands are named gossypol. - small issue but it makes unnecessary mess. Please do avoid thoughts shortucts. It is utterly important to make everything as plain and simple as it is possible especially in the beginning of introduction section.

[44] "Cotton pigment glands are found in the cotton plant" - I strongly recommend to delete this sentence or to restructure it. Till now I don't see the point of putting obvious statements as this one.

[56-57] Over-abundant language style - it should be changed into "..has two kinds of glands, internal and external."

Morphological observation of gland development in cotton plants:

[69-70] Please do restructure first sentence. As a readers we already know how these particular glands are called. Introduction part already widely described them.

[73-74] - In Gossypium barbadense, the cultivated species with the most gossypol glands. - I don’t understand the meaning of this sentence. Please do correct or erase it.

[68-146] – Lack of any reference. This whole part is undoubtedly interesting. Still, without a reference it is hard to tell for a reader if presented data is a common knowledge or the observations made by authors. If it’s

[78-79] [79-146] „We will study..” you should have used past tenses. For example „We studied..” still if this article ment to be review with own research/observation parts, it would be much better to add more details about the research/observation objects. After reading whole manuscript I still don’t know much about observed cotton. Is it yours observation? Is it something well-known already or is it something innovative? There are no methodology so how can I tell if its replicable? And it should be at least discussed.

Role of gland in cotton and its importance.

[179-184] „The relationship between cotton glands and gossypol production has been studied extensively” if that so why authors didn’t reference it anywhere? In this part there are some interesting assumptions, but without any reference it looks like like basic opinion. It should be corrected.

The Biosynthetic Pathway of Gossypol and Associated Terpenoids

[186] Please do correct citation numbering. Nineteenth reference is located far after 24,25,26 th. I can’t understand why is that. It makes reference checking far more difficult.

[191] As above. First use of eleventh citation is put between 19th and 29th. It also made its presence at line 226. There is chronological mess with citations here.

[196-202] Double citation „30” of the same work in two adjacent sentences.

[219-223] As above. Lack of citation order and double citation in the same reference.

Molecular Cloning of Genes Associated with Gossypol and Glands

[261-297] Please double check reference order.

Despite some minor flaws, if authors apply indicated corrections, I am eager to recommend this paper for publishing . It is worth mentioning that this manuscript introduces well-needed approach in gossypol-related review articles.

Reviewer 2 Report

Dear editor;

The current review gives recent knowledge and literature for the cotton gland morphogenesis and its transcriptional regulation with gossypol production. Especially genes associated with gland and gossypol production pathways were given in details in the review.

Despite the need of such kinds of reviews in the literature, the current one does not meet the critical criteria for a readable and attractive review.

First of all, the language of the article is not easy to understand and fallow. For instance, the first sentence of the abstract is starting with this statement: ‘Cotton, genus Gossypium L. containing many species, is a leading fiber and potential food crop, and cottonseed has performed satisfactorily in comparison to other conventional protein sources in various human nutritional tests’. I don’t understand what are the human nutritional test, conventional protein source, and cotton performance. The review full of such kinds of sentences that make impossible to understand and fallow the manuscript.

Secondly, the article was summarized the literature that was published from 1918 to 2018. Most of these literatures belong to 1990s and 2000s, which are too old for the rapidly developing and changing information era.

There are many recent studies on the cotton and gossypol production most of which was not mentioned in the article. Especially genome-editing based recent studies have important results in terms of cotton, gland formation and gossypol production. These studies also comprise a future perspective for the article.

Another problem is the utilization of unnecessary pictures (for instance, Figure 1) in the article.

Due to all these problems in the article, it is not possible to accept it without a major revision.

Reviewer 3 Report

I read the draft by Jan et al. “An overview of cotton gland development and its transcriptional regulation”. This report shed light on various aspects of the accumulation of natural poly-phenol aldehyde gossypol in Gossypium species. As a derivative of the isoprenoid pathway, gossypol act not only as a phytoalexin but is also active against plant diseases, being toxic to humans and animals; hence, the importance here relates to the safe use of cotton as a source fiber and oil. The draft revise knowledge not only about metabolic pathways, but also developmental features related to plant secretory tissue in the form of gossypol glands, which accumulate large quantities of this compound. The manuscript is reasonable well written, presents an interesting applied scope, and covers topics in molecular biology, biochemistry and physiology; so I’m am happy to recommend publication.

Round 2

Reviewer 2 Report

Dear adear authors and editors;
in my previous evaluation, I mentioned the language problems, usage of non-necessary pictures, lack of recent genome editing-based studies on the article. All these problems still exist in this revised version, and I do not see much correction or progress. I just want to show you a sentence found in the abstract part. 
' Cottonseed has been shown to be an effective source of protein compared with other conventional protein sources in various human nutrition studies' 
What do you mean by 'effective source of protein'
What is the 'other conventional protein sources'
do humans included in a nutrient test like animals'
The article is full of such kinds of non-scientific statements and it is hard to understand most of the sentences. Due to all these deficiencies, it is not possible for me to accept the article for publication. 

Round 3

Reviewer 2 Report

The revised article is not good enough to be published in the journal. The authors did not take my comments into consideration.
